# Strength dependency of frequency–magnitude distribution in earthquakes and implications for stress state criticality

Satoshi Matsumoto [1] ✉, Yoshihisa Iio[2], Shinichi Sakai[3] & Aitaro Kato[4]

Earthquake size distribution is characterized by the "*b*-value" of the power law decay, which exhibits spatiotemporal variations. These variations are sometimes evident before a large earthquake. Understanding spatiotemporal variations is key to developing a model for large-earthquake generation. Previous studies have shown that changes in the *b*-value are caused by the tectonic stress regime. Furthermore, lab experiments have demonstrated the *b*-value dependency of acoustic emissions on the criticality of the failure condition. However, the factors controlling the *b*-values during natural seismic activity are unclear. In this study, changes in the *b*-value in small earthquake sequences are investigated, focusing on failure criticality. Based on our high-precision focal mechanism dataset, we conclude that the *b*-value decreases as it nears a critical failure condition, providing a physical explanation for the reduction in b-value before a major earthquake. Our findings elucidate fault failure models, facilitating improvement in earthquake alerts and disaster mitigation.

The Gutenberg–Richter (G–R) relation[1], a well-known relationship between the frequency of earthquakes and fault size, is characterised by magnitude ($\log_{10} N(M) = a + bM$, where $M$ and $N$ denote the earthquake magnitude and the cumulative number of earthquakes exceeding M, respectively, and a and b are constants). Numerous studies have reported the *b*-values for natural earthquake sequences across various regions, revealing spatial variations. Some studies have demonstrated a correlation between stress conditions and the tectonic regime[2] and depth range[3–7], suggesting that the *b*-value is inversely proportional to differential stress[8,9]. On the other hand, temporal changes in *b*-values before and after large earthquakes have been reported by several studies[9–14], indicating the potential of the *b*-value as an indicator of the stress state in the source region of a large earthquake. Laboratory experiments have been conducted using rock samples to study the characteristics of *b*-values. These studies have found that *b*-values decrease with an increase in differential stress[8,15,16].

In other words, the *b*-value decreases when approaching the criticality of rock failure, i.e., the differential stress is close to the fault strength.

The observations on the behaviour of the *b*-value before large natural earthquakes vary significantly, underlining the importance of investigating the relation between the *b*-value and mechanical conditions in the media involving seismic activity. The findings of the experimental studies suggest that the *b*-value of a small natural earthquake sequence may decrease according to the state of stress and strength required for large earthquake genesis.

In particular, these results highlight the importance of considering the strength of the medium for estimating the *b*-value because, in rock fracture experiments, the differential stress may not often continue to increase until failure, but rather it may plateau just prior to failure. This implies that small earthquake activities exhibit fluctuation through a reduction of rock strength caused by factors such as a decrease in internal friction and an increase in pore pressure.

[1]Institute of Seismology and Volcanology, Faculty of Sciences, Kyushu University, Shimabara, Japan. [2]Disaster Prevention Research Institute, Kyoto University, Uji, Japan. [3]Interfaculty Initiative in Information Studies, The University of Tokyo, Tokyo, Japan. [4]Earthquake Research Institute, The University of Tokyo, Tokyo, Japan. ✉e-mail: matumoto@sevo.kyushu-u.ac.jp

The following inferences can be made from the above laboratory experiment results: 1) Seismic activity in a relatively strong medium that, under a higher differential stress, yields a relatively low b-value. 2) Proximity of the activity to the maximum strength of the medium reduces the b-value. Proximity implies criticality that the activity is under a condition close to critical for generating a large earthquake. Note that "differential stress" means loaded differential stress on a volume (or rock sample) and is not local differential stress at parts that might show spatial variation due to existing faults and heterogeneity. The difference between 2) and 1) is that 2) reveals that the b-value depends not only on the absolute value of differential stress but also on the proximity of differential stress to the failure condition. The observed b-value change towards large earthquake occurrence can be explained by either or both of the factors. Thus, it is difficult to distinguish between the two factors causing changes in the b-value. In particular, the latter factor has never been separately investigated in natural earthquake activity despite being implied in the b-value decline before a large earthquake. This might be caused by a lack of information about the strength of each earthquake fault among the seismic activity, which is used for measuring the b-value.

Under a stress condition loaded onto a medium, the state of stress on a various oriented fault plane in the media can be presented using a "Mohr circle"[17]. Normal and shear stress on any fault plane under the stress loaded onto a medium can be projected inside the circle. According to the Coulomb failure criteria (CFF) as expressed by the following formula[17], slip occurs when the shear stress on a fault reaches the strength ($\tau$) expressed by the cohesive strength ($\tau_0$), normal stress ($s_n$) friction coefficient ($\mu$), and pore fluid pressure ($p$).

$$\tau = \tau_0 + \mu(s_n - p) \tag{1}$$

For all earthquakes, the fault satisfies the criteria at the time of failure, although CFF is used for both stick-slip and stable sliding. We considered a volume containing many pre-existing faults on which triaxial stresses are loaded. Under this uniform stress condition, spatial variation in the earthquake fault plane direction indicates internal strength variation in the medium. The position plotted inside the Mohr circle, which is defined by normal and shear stress on a fault, reflects the strength of the fault. Thus, variations in the b-value across the Mohr circle can provide information on the relationship between the fault strength and b-value, excluding the effect of absolute stress magnitude.

To determine a point on the Mohr circle defined by the shear and normal stresses, the loaded stress tensor on the medium and the fault orientation of an earthquake have to be determined. The stress field in an area, excluding its absolute value, can be estimated using focal mechanisms or seismic moment tensors[18–21] based on the Wallace–Bott Hypothesis[22,23], which states that a fault slips in the direction of maximum shear stress. The principal directions of the stress tensor and stress ratio between the eigenvalues can be estimated using these methods, allowing us to draw a normalised Mohr circle for the area and determine the location of the focal mechanism data on the circle. We can discuss frequency–magnitude relations in different parts of the circle for numerous focal mechanism solutions in various areas on a unit Mohr circle.

In this work, in order to compensate for the lack of information about the fault strength in b-value evaluation, we analyse natural earthquake data and detailed discuss the relationship between the b-value and fault strength that is defined as a function of deviation from optimal fault plane directions expected by the regional stress field. Moreover, we suggest the unique dependency of the b-value on the fault failure condition.

## Results
### High-precision focal mechanism observations
To examine the dependency of the b-value on a normalised Mohr circle, we need to accurately obtain sufficient focal-mechanism data from a high seismicity area. Therefore, we selected the aftershock activity of the 2000 Western Tottori Earthquake of magnitude 7.3 on October 6, 2000, which is amongst the best-studied intraplate earthquakes, with detailed investigations into fault geometry[24], stress field[25–27], and structure[28]. Note that we here investigate a dependency of the b-value on fault strength among the aftershock activity without focusing on a spatial-temporal change in the b-value associated with the 2000 M7.3 event. Nevertheless, the relation between b-value and strength might provide implications for proximity to failure of further coming large earthquakes. To augment focal-mechanism data, we took advantage of using two datasets recorded by a dense seismic network during different periods deployed in the aftershock area (approximately 35 km in diameter) (Supplementary Fig. 1 & Supplementary Table 1). The first dataset, hereafter called 2000 OBS, was obtained from a seismic network comprising 59 three-component seismometers, installed just after the 2000 M7.3 rupture (13 October 2000–November 2000)[29]. The second dataset, hereafter called 2017 OBS, was obtained from an ultra-dense seismic network consisting of over 1000 vertical seismometers with a natural frequency of 2 or 4.5 Hz deployed from March 2017 to April 2018[28,30]. In addition, three-component seismograms at approximately 20 stations by Hi-net, which is operated by the National Research Institute for Earth Science and Disaster Resilience (NIED), were merged into the 2017 OBS.

To estimate the stress field, we determined the hypocentres and focal mechanisms of numerous small earthquakes using both the 2000 and 2017 OBS datasets (Fig. 1). The focal mechanisms of both 2000 and 2017 OBS were divided into spatial bins (Supplementary Fig. 2, 3). Then, we determined the relative stress tensors using the focal mechanism solutions within spatial bins, by employing a stress tensor inversion (Supplementary Fig. 2 and 5) (see "Methods"). The obtained results are similar to those of previous studies[25–27]. Subsequently, we obtained normalised shear and normal stresses for selected individual earthquakes of 2017 OBS (see "Methods") based on the fault plane and the relative stress tensor within the spatial bin containing the earthquake. The accuracy of two normal vectors of nodal planes of focal mechanisms used in the b-value estimation (2017 OBS) is approximately 11.6° in the magnitude range of −0.3 to 2.0 (Supplementary Fig. 4).

To estimate the b-value, we employed only 2017 OBS data because the magnitude range in 2000 OBS is too small to capture frequency distribution for estimating a stable b-value. Figure 2a shows the hypocentre distribution of 2017 OBS as recorded by the automatic detection system, adopted for b-value analysis (see "Methods").

Although previous studies have shown a dependency of b-value on depth (i.e., on differential stress: $\Delta\sigma$), the depth-dependent b-value behaviour is not clear in this study, as shown in Fig. 2b. We calculated the b-values for three depth ($Z$) ranges ($Z \leq 7.5$ km, $Z > 7.5$ km, and all depths) using ZMAP 7 software[31]. The b-value estimation using ZMAP 7 with a fixed minimum magnitude ($M_c = -0.3$, as adopted in the subsequent section) did not yield any considerable difference for the three depth ranges. This finding indicates that its depth dependency might be small (<0.1), even though the differential stress is expected to increase with depth[7]. According to an empirical relation[7] between b-value and differential stress, the relation between b-value and depth for a strike slip-stress regime ($\Delta\sigma = 20$ MPa/km) can be written as:

$$b = 1.23 - 0.0012\Delta\sigma = 1.23 - 0.024 \cdot Z \tag{2}$$

By applying this formula to the studied area, we expect that a b-value at a depth of 10 km should be approximately 0.1 higher than that at the depth of 5 km (i.e., average depths correspond to ranges of

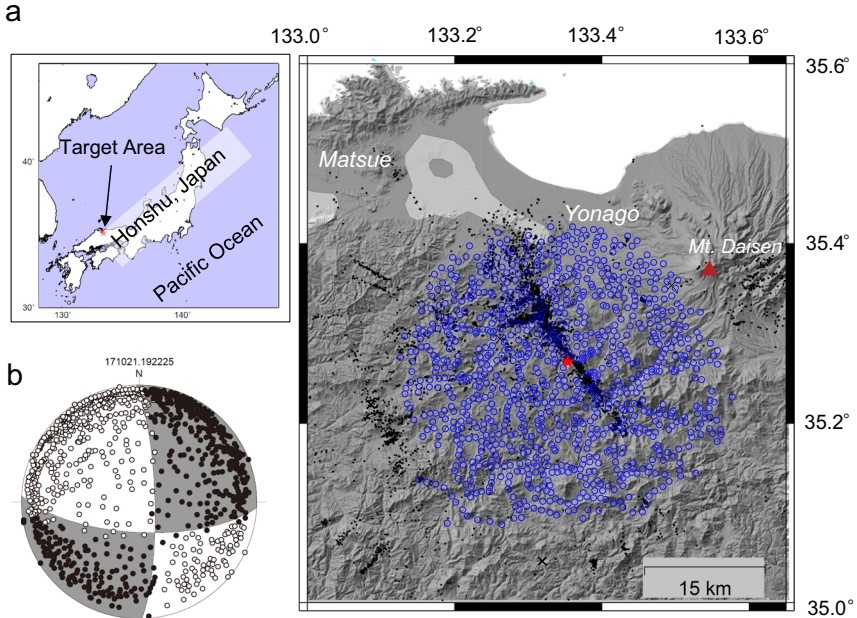

**Fig. 1 | Dense seismic observation and detected events. a** Seismic station geometry for 2017 observation, deployed in Southwestern Honshu, Japan, as shown in the index map. Blue open circles and dots plotted on the topographic map indicate seismic stations and observed hypocentres. The red star shows the epicentre of the 2000 Western Tottori earthquake (Magnitude 7.3). A solid triangle indicates the Daisen volcano. Topographic map is referred from Geospatial Information Authority of Japan. (https://maps.gsi.go.jp/development/ichiran.html) **b** The lower hemisphere plot shows an example of a focal mechanism solution for an event on 21 October 2017. The polarity of the first P wave motion of the event is plotted in the lower hemisphere. Solid and open circles indicate positive and negative polarities, respectively.

$Z \leq 7.5$ km and $Z > 7.5$ km, respectively). This indicates that another explanation other than the depth-dependency of the $b$-value must be considered if the difference in the $b$-value exceeds 0.1. Therefore, we consider the change in the $b$-value within the unit Mohr circle to reflect its dependency on fault strength related to the criticality of failure of earthquake faults.

### b-value map on a normalised Mohr circle

For the selected fault planes from the observation dataset (see "Methods"), normalised shear and normal stresses are plotted on a unit Mohr circle, as demonstrated in Fig. 3a. We estimated the $b$-value distribution by segmenting the Mohr circle (Fig. 3b). The lowest $b$-value was identified to be approximately $90° > \theta > 50°$ clockwise from the horizontal line at the outermost part of the circle. Figure 3c presents the $b$-value along with the standard error for the segments on the Mohr circle. The $b$-values for all data and broader segments ($r < 0.8$, $\theta = 0–90°$, and $\theta = 90–180°$) are also plotted for reference. The frequency–magnitude plots are shown in Fig. 4. The $b$-value is significantly smaller at $90° > \theta > 50°$ with $r \geq 0.8$ compared to the other ranges. The slip vector of the fault plane oriented to the plane by maximum and minimum principal stresses (σ1–σ3 plane) belongs to the high r segment, whereas that oriented to the oblique-slip of σ1–σ3 plane occurs in the low r segment. The high $b$-value observed in the low $r$ value segment suggests that it might be difficult for the rupture to grow when the slip vector of fault planes is oriented obliquely to the σ1–σ3 plane; subsequently, the earthquake magnitude gets suppressed, resulting in a higher $b$-value.

To interpret the $b$-value distribution on the Mohr circle in detail, we introduced a model where the proximity to CFF controls the b-value. The b-value reaches its minimum around the angle $\theta$, at which the Mohr circle touches a line defined by the CFF criteria (hereafter called the optimal range). A schematic of the Mohr circle in absolute stress dimension is shown in Fig. 5a. CFF with a minimum fluid pressure is plotted by a solid line. An example of proximity distribution (blue scale) is shown in the figure. The low $b$-value angle range, $\theta > 50$ shown in Fig. 3c, corresponds to a frictional coefficient of $\mu < 0.84$, considering that the touching point of CFF to the Mohr circle exists in the angle range. This range falls within or smaller than the range of internal friction coefficients of rock[32]. We could not resolve the range of friction coefficients adequately. However, the results revealed that a larger earthquake occurred in this state (small $dp$, which is the distance from the CFF line, in Fig. 5a). This suggests that relatively larger fault slips occur for cases with minimal contribution from the fault-strength-weakening mechanism from the optimal range, such as rising fluid pressure. In other words, in a large-magnitude earthquake, a fault plane optimal for a stress field is prone to rupture compared to unfavourably oriented fault planes. It results in the observed low $b$-value seismic activity for the optimal fault planes. This is reasonable as the releasable elastic energy on a fault plane is larger at the optimal range in the Mohr circle than outside the range under the same stress conditions. Fault planes plotted outside the optimal range may generate smaller earthquakes than those inside the range because larger earthquakes require large fault zones to be weakened, which is generally considered more difficult.

Our results showed that the weak faults plotted outside the optimal range were activated. The activation, attributed to pore fluid pressure, demonstrates that this area is under a partially high-pressure condition that can be ruptured by the unfavourably oriented fault to the stress field[33].

We considered that the proximity of a large earthquake depends on the distance to the line defined by CFF. For example, Fig. 5b shows $b$-value variation with dp, which is plotted along the horizontal axis, instead of the angle from −σn, as shown in Fig. 3c, by assuming $\mu = 0.6$. In Fig. 5b, the $b$-value seems to increase with increasing $dp$. However, the variation is not fully explained by the above simple model, suggesting a complex condition, such as the heterogeneous distribution of friction coefficient.

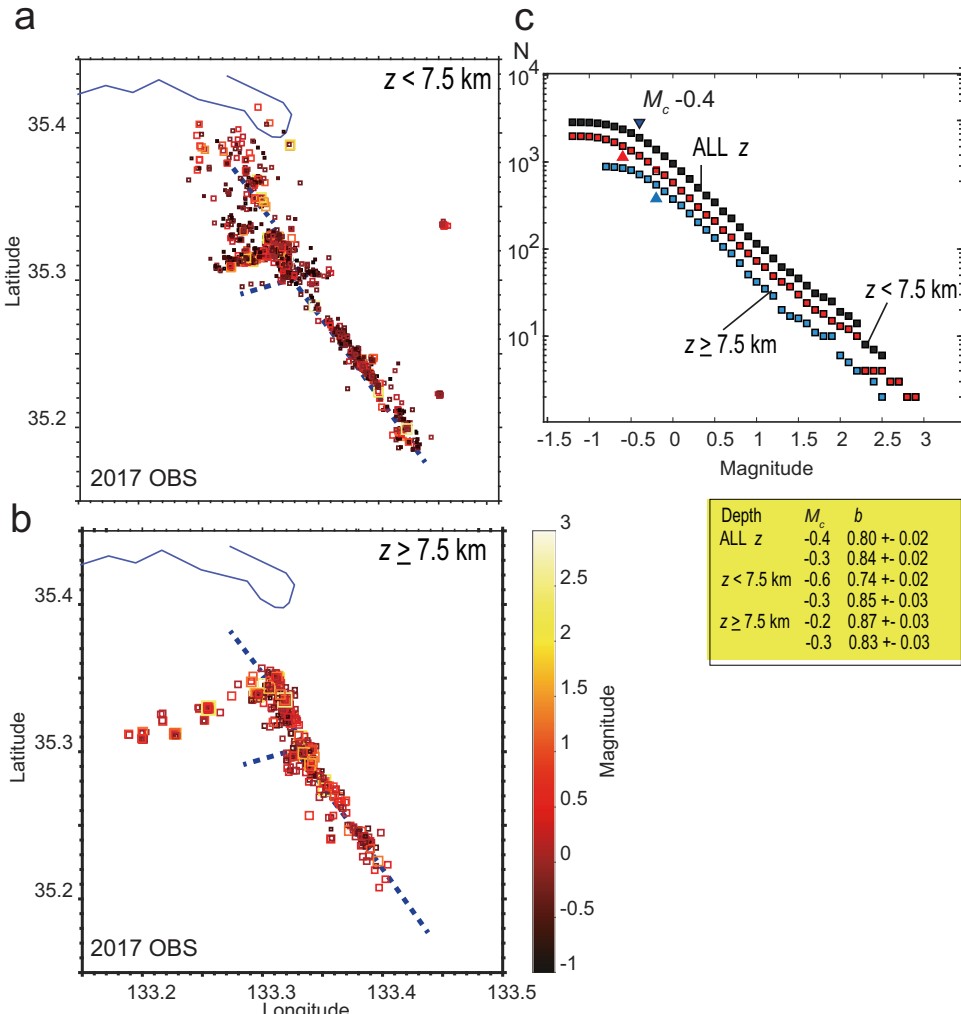

**Fig. 2 | Analysed events and their frequency magnitude distribution.**
**a** Hypocentre distribution shallower than depth of 7.5 km used in *b*-value estimation (coloured square). The stress tensors in the spatial bins were estimated using the plotted events as described in the Methods section. Coloured symbols, which are colour-scaled for magnitude, indicate events with focal mechanism solutions adopted in the following section. The blue dashed line indicates the co-seismic fault trace. **b** Same as a except depth range over 7.5 km. **c** Frequency–Magnitude distributions for events at all depths, depth range shallower than 7.5 km, and equal to and deeper than 7.5 km. The *b*-values are shown at the bottom. The values were estimated for automatically determined $M_c$ and fixed $M_c = -0.3$.

The absolute *b*-value was not a concern in this study because it may change in different regions. In general, the effects of absolute differential stress on the b-value can be included in the results of this study, but the effect was not significant, as estimated from the depth dependency of *b*-values.

We revealed how the *b*-value depends on the fault strength based on the highly precise dataset of the focal mechanisms and hypocentres. Regarding an earthquake catalogue without any focal mechanism solution, the change in the *b*-value can be smaller than that observed in the present study, as earthquakes may occur along various fault geometries. However, a lower *b*-value indicates that the fault planes of the earthquakes in the catalogue are under a critical stress state for the failure criteria rather than under activation by weakening fault strength.

In the present study, we investigated the *b*-value dependency upon fault strength. Our results show that the earthquakes with the lowest b-values are under a stress state close to critical failure conditions (high proximity). In other words, the *b*-value decline before large earthquake occurrences reported by previous studies can also be interpreted as the change in the b-value is related to the proximity of earthquake generation under Coulomb failure conditions. This reveals

the importance of monitoring *b*-value dependency on fault orientation as it has the potential to help us evaluate the proximity of a large earthquake.

## Methods
### Focal mechanism estimation
We determined the deviation of slip direction from the optimal fault slip direction expected from the stress field. The study area was divided into spatial bins as shown in Supplementary Fig. 2. For the 2017 OBS data, we applied automatic picking and focal mechanism determination processes to the continuous seismic records after checking the quality of the extracted 100 event data by comparing the manually picked and automated results[30]. Similar to a previous study[34], we adopted a smoothed, depth-dependent velocity structure for hypocentre determination. Only the events within the area covered by the 2017 OBS network were considered, containing a total of 3,641 events[30]. The 2,828 events included in the bins are extruded for subsequent analysis. For 2000 OBS, we used 2,285 events after manually picking the polarity[29]. P-axis distributions for both datasets are shown in Supplementary Fig. 3 (P-axes of events included in the spatial bins where stress field estimated are emphasised). The focal mechanisms of

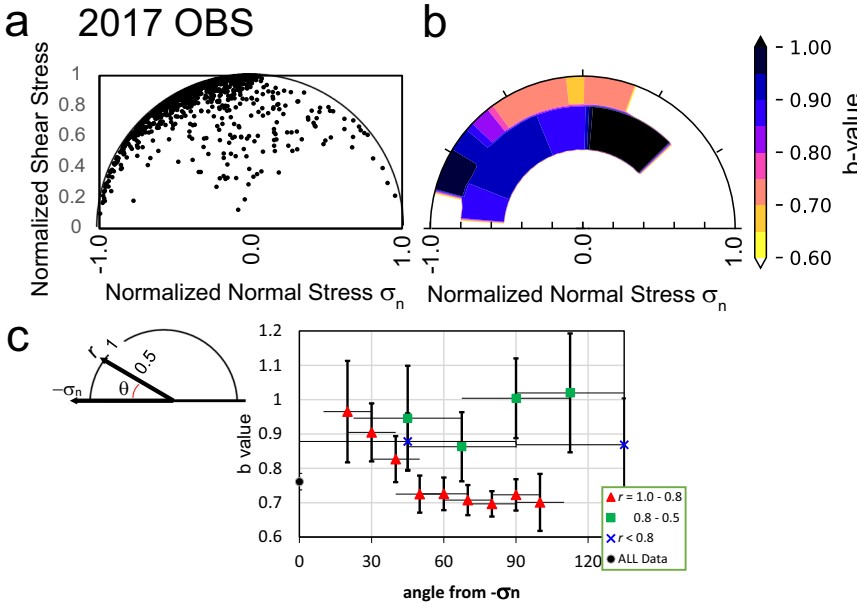

**Fig. 3 | *b*-value dependency on the fault strength. a** Shear and normal stress distribution on a unit Mohr circle. Data plotted are events contained in spatial bins with stress tensor resolved. **b** *b*-value distribution on the unit Mohr circle. The colour in the circle indicates *b*-value as shown in the colour scale. **c** *b*-value for the angle θ from -σ_n defined in the upper inset. Vertical and horizontal bars indicate the standard error of the estimation and angular range on the area of the Mohr circle for estimating the *b*-value, respectively. The range of radius for the estimation is indicated as r. The upper inset shows the definition of coordinates on the Mohr circle.

the selected data were determined using the HASH algorithm[35]. The completeness of the detection process reached an M value of approximately −0.5 and 1.7 for 2017 and 2000 OBS, respectively, by ZMAP analysis. High-quality focal mechanism solutions were obtained. For each event, the optimal solution was derived as the tensor average from all possible focal mechanism tensors. We also generated 1000 focal mechanisms for each event based on the solution distribution using HASH. The datasets were utilised in the subsequent processing. We created 1000 datasets of focal mechanism solutions by random sampling of the above-mentioned 1000 focal mechanism solutions. This dataset was used to calculate the confidence range of parameters by bootstrap random sampling procedure. The standard errors in the normal vectors of the nodal planes of the events are plotted in Supplementary Fig. 4, which reveal the average values of 10.8° and 19.8° for the 2017 and 2000 OBSs, respectively.

## Stress field

The normalised deviatoric stress tensors were estimated for the spatial bins. The tensor was derived from the seismic moment tensor data of the aftershocks, utilising a method[18] based on the equivalent relation between the inelastic strain and the released seismic moment tensor[36]. We employed the stress field estimation using both 2017 and 2000 OBS data. We created a moment tensor dataset based on the focal mechanism and earthquake magnitude ($M_a$). The tensor shape and scalar moment ($M_O$) were obtained from the estimated focal mechanism and transformed value from $M_a$ through an empirical relationship used in a previous study[27]: (log ($M_O$) = 1.15 $M_a$ + 10.548). Notably, stress tensor estimation is independent of the fault plane selection, as it utilises moment tensor data. The stress tensor estimated from the seismic data provides a normalised tensor shape characterised by the principal directions and ratio among the eigenvalues. The deviatoric stress tensors were estimated in spatial blocks, set at intervals of 3 km horizontally and at 2.5 km depth (Supplementary Fig. 2). In addition, we performed the same procedure for the block distribution shifted by half a block to the north and east from the origin to smoothen the results. The principal directions and relative

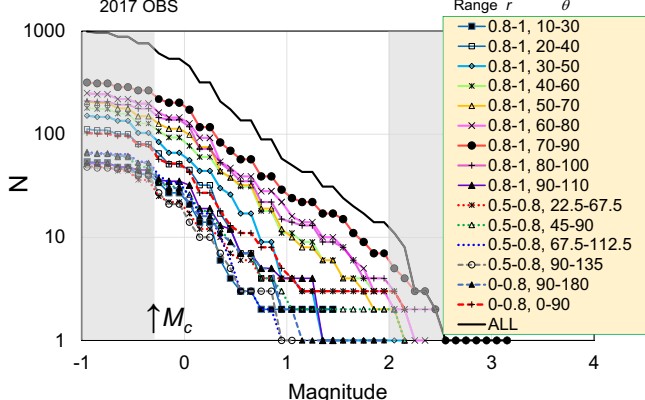

**Fig. 4 | The cumulative number of events for the magnitude of the earthquake.** The plotted lines are curves for estimating the *b*-value shown in Fig. 3c. Each line corresponds to the curve for each r and θ range on the Mohr circle, as shown in the legend. The magnitude range for the estimation, which is shown by the unshaded area, is from $M_c = -0.3$ to $M = 2$. Lines with solid circles show a frequency–magnitude curve taking minimum *b*-value (i.e., θ = 70–90, r = 0.8–1).

values of the eigenvectors that revealed the normalised deviatoric stress tensor were estimated from the events in a spatial bin. The focal mechanism data and the estimated parameters in each spatial bin within the 95% confidence range are shown in Supplementary Data 1. The results obtained using the 2000 OBS data coincide with those using the 2017 OBS data within the 95% confidence range (Supplementary Fig. 5). Therefore, we adopted both datasets for estimating parameters. The results align with those of previous studies[25–27]. The use of both datasets suppressed the confidence range and increased the number of available spatial blocks for the subsequent analysis.

The parameters with substantial accuracy were mainly limited to spatial bins around the aftershock area with relatively high seismicity. In the subsequent discussion, we focused solely on the data within the spatial bins exhibiting significant accuracy, as shown in Supplementary

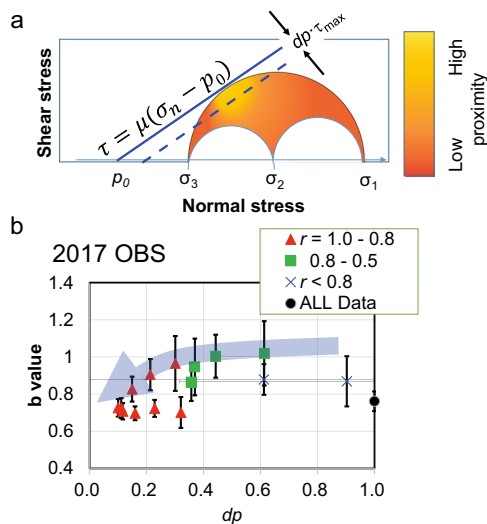

**Fig. 5 | Dependency of $b$-value on strength. a** Schematic illustration of the proximity on a Mohr circle in the absolute stress domain. Proximity are plotted by colour scale shown in the right. The highest of the proximity is located at a point touching the Coulomb failure criteria (CFF) segment (blue solid line) to the Mohr circle. **b** $b$-value variation same as Fig. 3c except horizontal axis in $dp$ (distance from CFF segment as shown in (a) for a case assuming a friction coefficient of $\mu = 0.6$. Vertical bars indicate the standard error of the $b$-value estimation.

Fig. 3. The adopted magnitude range for $b$-value estimation was narrow for 2000 OBS (approximately $1.7 < M < 3.0$, satisfying the G-R relation as shown in Supplementary Fig. 6). Finally, we analysed 2,155 (1,412 for $M_c = -0.3$) focal mechanism data from 2017 OBS to investigate the strength dependency of the $b$-value.

A comparison of the number of focal mechanisms with hypocentres, including spatial bins, is shown in Supplementary Fig. 7 as the frequency–magnitude distribution. The number of focal-mechanism data points was slightly lower than that of the hypocentre data. This difference can be attributed to a less favourable geometrical condition of the station distribution or a lower signal-to-noise ratio of the seismograms used to determine the focal mechanism. The estimated $b$-values exhibit a discrepancy of approximately 0.07 owing to the large difference in the small magnitude range. However, no specific focal mechanism was identified in this dataset. Therefore, the relative $b$-value changes on the parts of the Mohr diagram are discussed.

### Shear and normal stress estimation for stress tensor and selection of fault planes

The normalised shear and normal stresses were calculated from the fault geometry and stress tensor ($\sigma$). According to a previous research[17], normal stress vector $\sigma_N$ is a product of traction vector **T** and unit fault normal vector **n** [$\sigma_N = (\mathbf{T} \cdot \mathbf{n})\mathbf{n}$], where the traction vector is $\sigma$ applied to **n**; $\mathbf{T} = \sigma\mathbf{n}$. The shear stress vector $\tau$ is expressed as $\tau = \mathbf{T} - \sigma_N$. From the seismicity data (focal mechanism or moment tensor), we can only determine the normalised deviatoric stress tensor $\hat{\sigma}$, characterised by the principal directions of the eigenvalues and the ratio between the eigenvalues. This implies that the magnitude of $\sigma$ is arbitrary; therefore, the relative relationship between the shear and normal stresses was obtained. In this case, the relationship provides $\sigma_N/|\mathbf{T}|$ and $\tau/|\mathbf{T}|$ from $\hat{\sigma}$ and **n**. Therefore, we can define a Mohr circle with a unit diameter, with its centre point at the origin of the coordinates. For instance, the maximum shear stress under the stress tensor acting on a plane with a normal vector that rotates 45° from the maximum principal direction to the minimum principal direction of the stress tensor is plotted at (normal stress, shear stress) = (0, 1) on the normalised (unit) Mohr circle. In this study, we used coordinates to

express the location on the Mohr circle by $r$ (radial length from the centre, i.e., ranging from 0 to 1) and $\theta$ (clockwise angle from the horizontal).

To estimate the normal and shear stresses on the unit Mohr circle, a fault-normal vector is required. We have two candidates for the normal vector for each event from the focal mechanism solution owing to the existence of two nodal planes. The following scheme was used to select one of the two planes. First, we eliminated the fault plane if it had a misfit angle ($d\theta$), which is the discrepancy between the slip vector of the focal mechanism and the maximum shear stress direction expected from the stress field, exceeding 30°. We set two conditions for fault plane selection: (1) misfit angle ($d\theta$) and (2) comparison lengths of segments ($dp$) to the line defined by CFF for a certain friction coefficient ($\mu$) from two points calculated by the normal vectors of the candidates. A schematic of this process is shown in Supplementary Fig. 8. Shear strength in CFF is expressed in (1) and $p \propto dp$ represent relation that pore fluid pressure $p$ is proportion to $dp$, respectively. A point touching the CFF line on the unit Mohr circle with angle $\theta_c$ can be expressed by $\theta_c = \tan^{-1}(1/\mu)$, as shown in Supplementary Fig. 8b and reported by Jaeger et al.[17]. Unfavourable fault planes for the stress field require a high value of $p$ to generate rupture, which is more challenging to achieve naturally than in situations with a low $p$, owing to difficulties sustaining a high $p$.

The $d\theta$ range for each plane was calculated from the stress fields within a 95% confidence interval in the spatial bin containing the event. Initially, we selected a plane with a smaller $d\theta$ range than the other plane (condition 1). However, if the $d\theta$ ranges overlapped, we chose a plane with a smaller $dp$ (condition 2). This criterion is known as the selection based on fault instability, as defined in a previous study[37]. To calculate $dp$, we set $\mu = 0.6$, the typical internal friction coefficient[27]. However, assuming $\mu = 0.2$ or 1.0 did not change the result (Supplementary Fig. 9).

After completing the above process for each event, we compared the selected fault plane for an estimated stress tensor with that for another one. As described in the stress field section, spatial block distribution shifted half the block size horizontally, meaning two candidates of the stress field exist for most events. We used a fault plane and a stress field combination that provided a relatively small misfit $d\theta$.

### $b$-value estimation

The $b$-value, which represents the power-law decay of the frequency–magnitude distribution, has been estimated for various regions and exhibits spatial and temporal variations. We followed the standard method (i.e., maximum likelihood method) to estimate the $b$-value[38–40]. We set a consistent magnitude range (minimum and maximum magnitudes) for each part of the Mohr circle, as comparing the $b$-values across different parts of the circle is crucial for this study. We used the magnitude range from −0.3 to 2, with the lower bound determined using ZMAP software[30]. The upper bound was set at a position at which the segment of the frequency–magnitude relation slightly deviated from the linear relation between log $N$ and $M$ (i.e., off from the Gutenberg–Richter relation). Additionally, the number of earthquakes above the upper bound was too small for analysing the $b$-value variation within the Mohr circle.

We estimated the $b$-values for the parts of the Mohr circle that contained over 50 events, excluding events close to the co-seismic fault of the 2000 Western Tottori earthquake. The heterogeneity of the stress field may lead to improper estimation of the fault direction relative to the stress field. Based on the co-seismic fault model[24], we calculated the distance to the fault planes and selected events that were located farther than a certain distance ($R$). In this study, we set $R = 0.5$ km. The results for different distance settings are shown in Supplementary Fig. 10a. Increasing $R$ significantly decreased the number of usable events due to the concentration of hypocentres

around the co-seismic fault planes (Supplementary Fig. 10b). The number of excluded events with larger misfits, as described in the previous section, decreased with increasing $R$, indicating the influence of the nearby faults on events. We selected $R = 0.5$ km, which led to the utilisation of 65% of events from the initial set, while 8% were excluded.

The unit Mohr circle was divided into parts expressed by range ($r$ minimum, $r$ maximum; $\theta°$ minimum, $\theta°$ maximum) in the coordinate shown in the inset of Fig. 2(c); the ranges are (0.8, 1; 10, 30), (0.8, 1; 20, 40), (0.8, 1; 40, 60), (0.8, 1; 60, 80), (0.8, 1; 70, 90), (0.8, 1; 80, 100), (0.8, 1; 90, 110), (0.5, 0.8; 22.5, 67.5), (0.5, 0.8; 45, 90), (0.5, 0.8; 67.7, 112.5), (0, 0.8; 0, 90), and (0, 0.8; 90, 180). Average and median values of standard error among the events belonging to each $r$ and $\theta$ part are listed in Supplementary Table 2. The errors in $r$ and $\theta$ were calculated from the optimal and 1000 focal mechanism dataset and the estimated stress field by the bootstrap random sampling procedure. Although the error range in $\theta$ became larger than that in $r$, the division in the Mohr circle was adequate to capture the characteristic of $b$-value distribution.

## Data availability
The focal mechanism data generated in this study have been deposited available in the Zenodo repository, 10.5281/zenodo.8242561. Phase data used in this study is accessible in the Zenodo repository, 10.5281/zenodo.11231063.

## Code availability
The HASH algorithm used to determine the focal mechanism is accessible at https://www.usgs.gov/software/hash-12. The ZMAP programme by Wiemer[26] is accessible at http://www.seismo.ethz.ch/en/research-and-teaching/products-software/software/ZMAP/). We plotted the figures using GMT5 (http://gmt.soest.hawaii.edu/projects/gmt) and Microsoft Excel 2016. The other customised codes used in this study are available in the Zenodo repository, https://doi.org/10.5281/zenodo.10910587, upon reasonable request to the corresponding author.

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

## Acknowledgements
We thank the staff and students of Kyushu and Kyoto Universities and the University of Tokyo who maintained the seismic stations for obtaining good-quality seismic data. We also used seismic data from the Japan Meteorological Agency and Hi-net.

## Author contributions
S.M. performed data processing and analysis of the micro-earthquakes, including relative strength estimation. S.M., Y.I., S.S., and A.K. managed and conducted hyperdense seismic observations. All the authors contributed to the discussion and approved the final manuscript.

## Competing interests
The authors declare no competing interests.
