## [Peer Review File · Nature Communications]

Strength dependency of frequency–magnitude distribution in earthquakes and implications for stress state criticalityEditorial Note: Parts of this Peer Review File have been redacted as indicated to remove third-party material where no permission to publish could be obtained.

REVIEWER COMMENTS

Reviewer #1 (Remarks to the Author):

Matsumoto et al.: Examining the strength dependency of frequency-magnitude distribution in small Earthquakes: implications for stress state criticality

Matsumoto et al investigate changes in b-value in an earthquake sequence in Japan using seismic network data. The authors are interested in understanding how/if b-value changes when the fault is in a critical state/close to failure. They derive stress state data from moment tensors and analyze changes in b-value in the context of stress state. The data, though hard to follow, are consistent with the idea that b-value decreases with increasing stress. Overall, I found that article difficult to follow for reasons pointed out below. I think the overall scientific question about b-value and stress state is interesting, but I'm not sure what additional insights one can gain from this study. For example, we have known for quite some time that b-value decreases (in some cases) leading up to the mainshock. The results here I believe are consistent with this idea? But beyond that I'm not sure what additional insights have been gained from this study. But maybe I'm missing something fundamental. I think the paper is worth publication after major changes have been implemented.

Major comments:

1. The details of the dataset/catalog that the authors use to derive the stresses and b-values is unclear. They mention that the focus is on 2000 M 7.3 earthquake, but the exact time and space windows used for the analysis is unclear which makes it difficult to fully understand the results/discussion. Figure 1A shows the station network but do not clarify what the spatial bounds are for area of interest. If the focus of this work is on stress state criticality then I would imagine that the events/analysis would focus on the seismicity leading up to the M7.3 event? But there is also mention of aftershocks in regards to the analysis which makes this even more confusing (L189).
2. In general, I find most of the plots hard to follow and lacking detail. See comments below. If the focus is on stress state criticality, then why not cross plot stress and b-value as a function of time leading up to failure/mainshock. And/or plot b as a function of stress state? Figure 3 shows some of these details, but it doesn't address criticality/close to failure part.
3. The current discussion about the implications of the results presented here is difficult to follow and it's unclear what the broader implications of this work are and how it connects and/or adds to the existing literature. For example, we have known for quite some time that b-value decreases (in some cases) leading up the mainshock and is connected to the stress state. It seems that this work is

consistent with this idea, but it's unclear what new information/insights has been gained from the current study.

Minor comments:

L42: Normal stress and differential stress dependence also observed in: Scholz, 1968; Rivere et al., 2018; b-value is also thought to serve as a potential "stress meter" (e.g., Tormann et al, 2012).

L44: I would say that the reduction in b-value has been observed before some large earthquakes. I don't think it's a robust characteristic for all events..if it is then maybe it's being masked by our inability to detect to small earthquakes. Should also cite: Gulia & Weimer, 2019; Nanjo et al., 2012;

L50-51: Bolton et al. demonstrate that b-value is a function of both stress and fault slip velocity. L55: What do you mean by initial robustness?

L57: The wording for #2 is hard to follow; I believe what you are saying is that b-value reduces when the fault is in a critical state/near failure (e.g., Scholz, 1968)? I'm confused when you say that this has never been investigated for natural earthquakes. In Gulia & Weimer, 2019 and Nanjo et al., 2012, they both see a reduction in b-value leading up to the mainshock. The fact that b-value is reducing before failure implies that the fault is in a critical state/near failure, correct? What about these findings is incomplete in terms of b-value and fault stress state?

L66: Important to note that frictional slip occurs when the stress surpasses the strength, but Mohr-Coulomb doesn't describe whether or not this slip will be unstable (i.e., earthquake) or stable (aseismic creep).

L68: Can you please explain what you mean by this statement: Under a uniform stress condition...

L91: Does the hypocenter area correspond to the ring of stations in Figure 1? Is this what you mean by "target area on lines 107, 189? Please clarify.

L94: What are the temporal bounds of the catalog used in this study? I'm confused because in lines 91 you say observations in the hypocenter area of the 2000 Western Tottori earthquake which leads me to believe that the spatial/temporal window of this study is centered around the 2000 mainshock. But here on line 94 you state the yearlong observations in 2017. Please explain.

L97-99: Why did you use a depth cutoff at 7.5 km? Figure 2 could be improved by making the symbols in panels a and b bigger, increasing the font size on the figures, and labeling the F-M curves in panel B.

L102: How do you calculate M_c ?

L106: This should be formatted as an equation as opposed to have the equation in-line. I would also make it clear that this equation is derived from Scholz, 2015 and not the data presented in this work.

Figure 3: How are the shear and normal stresses normalized? What does it mean to have a negative normal stress? How are the data in Figure 3a /3b segmented? How is Figure 3B subdivided? What do the numbers in the legend in Figure 3D correspond to? Do each one of these F/m lines correspond to the data in Figures 3B-3C. please clarify. Also, it would be useful to point out where M_c is located on these plots and the predicted F/M (i.e., fit) to the data.

L126: What does the range of radius, r , physically mean? And what does this imply about the relationship shown in Figure 3C. In other words, why does b when r is between 0.8-1, but is independent of normal stress between 0.5-0.5?

L122-128: Does this region correspond to a high normal stress? Can you say anything about where/how close the fault is to failure? This would help address the question of whether b -value is low when the fault is critically stressed.

L129: It would be useful to show a mohr-coulomb plot, the failure envelope and the distribution of b -values. Perhaps some of this information is shown Figures 3a-b?

L133-136: I do not follow this section. Can you please rephrase/explain what you mean here: "This suggests relatively larger fault slips..."

L144-145: Please explain what you mean here. Where in Figure 3 are the optimal and non-optimal planes that you are referring to?

L146: Why does pore-pressure need to be the causative agent for causing unfavorable planes to slip?

L171-186: What are the spatial and temporal bounds of the catalog used in this study? How many events are in the catalog? 3641? Is the network called "The 2017 Network" or is 2017 simply the year the network was deployed. Please clarify.

L242:-244: Can you provide an example and/or show in Figure A5 how the misfit angle is determined.

L222-265: What are the uncertainties associated with the normalized normal and shear stresses measured here?

L269: What do you mean by standard method? Maximum likelihood? How do you determine M_c ?

L277: Why do you exclude events close to the fault plane?

Reviewer #2 (Remarks to the Author):

Matsumoto et al. analyzed two well-recorded earthquake sequences in the Tottori region, Japan, focusing on variations in focal mechanisms and b-values. Variations in b-values are linked to relative differences in shear and normal stresses determined from focal mechanisms. The authors find that faults with shear to normal stress ratios close to a critical stress state exhibit lower b-values than faults that are not optimally-oriented for slip. The authors interpreted their observations in the context of previous lab results which reported decreasing b-value with increasing differential stress and proximity to failure.

The study is based on an unique dense-array dataset from Japan. The dense station coverage in the Tottori region, enabled the authors to compute of focal mechanisms of even tiny events, (apparently down to $M \sim -1$, Fig. A4). Robust focal mechanism estimates for such small events is generally not possible because (e.g. due to low SNR, unknown polarities, and large azimuthal gaps). Consequently, I believe publishing the dataset and observations would be of great value to the scientific community.

Nevertheless, the link between focal mechanisms, fault stresses and b-values is not described in sufficient detail. The authors should better demonstrate that they are indeed able to robustly constrain focal mechanisms and it would be beneficial to report uncertainties as a function of magnitude. The description of the two different earthquake sequences and respective seismic instrumentation could be more detailed, and the authors should add figures that clearly show which events were selected to invert for stress and b-values (Fig. A3 is not clear enough because it doesn't show the underlying events).

Overall, the results are interesting and novel, but I believe it will take significant revisions, including redrafting many figures to turn this into a more suitable manuscript.

Method:

- Explain and show in more detail which events were selected to compute regional stress tensor variations, magnitude distributions and b-values!

- Results from focal mechanism and stress tensor analysis of differently discretized datasets in 2000 and 2017 should be shown as maps. Fig. A3 is a start but the results are difficult to evaluate. Exemplary stress tensor maps, can be found in the following papers:

Yang, W., & Hauksson, E. (2013). The tectonic crustal stress field and style of faulting along the Pacific North America plate boundary in southern California. *Geophysical Journal International*, 194(1), 100–117. <https://doi.org/10.1093/gji/ggt113>

Martínez-Garzón, P., Bohnhoff, M., Kwiątek, G., & Dresen, G. (2013). Stress tensor changes related to fluid injection at the Geysers geothermal field, California. *Geophysical Research Letters*, 40(11), 2596–2601. <https://doi.org/10.1002/grl.50438>

Results:

- How does the resolved stress orientation compare to spatial variations in b-values? It would be nice to show both results in a side-by-side comparison together with a map of the hypocenters.

Presentation and Structure:

- The presentation of the results needs substantial improvements. The paper is poorly structured, and a lot of key information are scattered across the main text, Method section and supplement.
- Display items and descriptions need much improvement. Lack of clarity in writing and figures significantly subtracts from the otherwise well-conceived study and the unique dataset.

Uncertainties:

- Fig. 3 only shows uncertainty in b-value and stress. It would be helpful to also show how well focal mechanism of small events were resolved.

Minor Comments:

L 24: Grammar! First sentence needs to be rewritten.

L 43: explain the term: "inclination of differential stress" (differential stress is not a tensor!)

L 44: Decreasing b-values before large earthquakes are rarely observed. Note that the provided citation (Smith 1981) argued for an increasing in b-value before moderate earthquakes in New Zealand. These statements need to be much more nuanced to accurately portrait the current observations.

L 58: rewrite!

L 68: Grammar

L 72: rewrite!

L 90: Describe the seismic network and type of instruments in more detail!

L 92: Explain the two different earthquake sequences in 2000 and 2017/18 in more detail. The point was quite confusing because it was unclear what earthquake sequence was used to derive the primary observations and results.

Fig.1: Caption: Dayson volcano is not in the "lower left"

Writing and grammar need to be improved throughout.

We would like to submit our article revised based on comments by you and reviewers.

The reply for the comments is described as follows (blue colored sentences).

We hope this revision is sufficient to answer for the comments.

Reviewer #1 (Remarks to the Author):

Thank you for your valuable comments.

We answered your questions and comments by blue sentences as follows.

Major comments:

1. The details of the dataset/catalog that the authors use to derive the stresses and b-values is unclear. They mention that the focus is on 2000 M 7.3 earthquake, but the exact time and space windows used for the analysis is unclear which makes it difficult to fully understand the results/discussion. Figure 1A shows the station network but do not clarify what the spatial bounds are for area of interest. If the focus of this work is on stress state criticality then I would imagine that the events/analysis would focus on the seismicity leading up to the M7.3 event? But there is also mention of aftershocks in regards to the analysis which makes this even more confusing (L189).

We added explanation about seismic observations in two periods. We discuss b-value dependency on fault strength by using aftershock activity of 2000 Western Tottori earthquake. We did not discuss stress change before and after the M7.3 earthquake. We clarified this point in the revised version. (L106 – 109)

2. In general, I find most of the plots hard to follow and lacking detail. See comments below. If the focus is on stress state criticality, then why not cross plot stress and b-value as a function of time leading up to failure/mainshock. And/or plot b as a function of stress state? Figure 3 shows some of these details, but it doesn't address criticality/close to failure part.

We did not study about temporal variation in b-value. As you pointed out, temporal variation is useful to discuss the criticality of failure condition. At the same time, it is

difficult to separate stress build up and criticality from b-value change from temporal variation. Therefore, we focus b-value variation in fault strength to capture criticality.

3. The current discussion about the implications of the results presented here is difficult to follow and it's unclear what the broader implications of this work are and how it connects and/or adds to the existing literature. For example, we have known for quite some time that b-value decreases (in some cases) leading up the mainshock and is connected to the stress state. It seems that this work is consistent with this idea, but it's unclear what new information/insights has been gained from the current study.

The point in our study is b-value decrease for fault plane oriented favorable direction to the loaded stress on the medium. This seems to be natural but no study exist from the point of view about fault strength in natural earthquake activity.

Minor comments:

L42: Normal stress and differential stress dependence also observed in: Scholz, 1968; Rivere et al., 2018; b-value is also thought to serve as a potential "stress meter" (e.g., Tormann et al,2012).

We added the references. (L.43 in revised version)

L44: I would say that the reduction in b-value has been observe before some large earthquakes.

I don't think it's a robust characteristics for all events..if it is then maybe it's being masked by our inability to detect to small earthquakes. Should also cite: Gulia & Weimer, 2019; Nanjo et. al., 2012;

We commented about opposite case of b-value change in Smith 1981 and adding references you pointed out. (L.45 in revised version)

L50-51: Bolton et al. demonstrate that b-value is a function of both stress and fault slip velocity.

Bolton et al. showed b-value dependency on fault slip velocity as you pointed out. However, we could not discuss about this point in the present analysis and excluded the point. They also demonstrated shear stress dependency disappear in low shear stress. It suggests that proximity of critical state for failure is a parameter for b-value change.

L55: What do you mean by initial robustness?

It means original strength without any weakening process such as rising pore pressure.

We changed the expression. (L.60-61 in revised version)

L57: The wording for #2 is hard to follow; I believe what you are saying is that b-value reduces when the fault is in a critical state/near failure (e.g., Scholz, 1968)? I'm confused when you say that this has never been investigated for natural earthquakes. In Guila & Weimer, 2019 and Nanjo et al., 2012, they both see a reduction in b-value leading up to the mainshock. The fact that b-value is reducing before failure implies that the fault is in a critical state/near failure, correct? What about these findings is incomplete in terms of b-value and fault stress state?

We would like to say b-value reduction occurs when stress state is critical. Previous studies demonstrated b-value decrease before large earthquake occurrence. On the other hand, b-value change due to change in depth and stress regime, reveals absolute stress level effect on b-value. These observations can be explained by inverse proportion of b-value to stress increasing without proximity to critical condition. Therefore, we attempted to discriminate the criticality effect from factors making b-value change.

L66: Important to note that frictional slip occurs when the stress surpasses the strength, but Mohr-Coulomb doesn't describe whether or not this slip will be unstable (i.e., earthquake) or stable (aseismic creep).

We changed sentences in L85 - 87 as follows;

According to the Coulomb failure criteria (CFF)¹⁷, slip occurs when the shear stress on a fault reaches the strength expressed by the cohesive strength, normal stress, friction coefficient, and pore fluid pressure. For all earthquakes, the fault satisfies the criteria at the time of failure although CFF is used for both stick and stable sliding.

L68: Can you please explain what you mean by this statement: Under a uniform stress condition...

We consider a volume containing many pre-existing faults. Tri-axial stresses are loaded onto the volume. (We added the sentence L87-89)

L91: Does the hypocenter area correspond to the ring of stations in Figure 1? Is this what you mean by "target area on lines 107, 189? Please clarify.

At first, we set target area covered by the stations in 2017 OBS. After stress tensor inversion, we adopt only events in spatial bins where stress tensor can be estimated. We described about this point more detail.

L94: What are the temporal bounds of the catalog used in this study? I'm confused

because in lines 91 you say observations in the hypocenter area of the 2000 Western Tomori earthquake which leads me to believe that the spatial/temporal window of this study is centered around the 2000 mainshock. But here on line 94 you state the yearlong observations in 2017. Please explain.

We rewrote the description about temporal observations more detail in “High-precision focal mechanism observations”.

L97-99: Why did you use a depth cutoff at 7.5 km? Figure 2 could be improved by making the symbols in panels a and b bigger, increasing the font size on the figures, and labeling the F-M curves in panel B.

We showed this figure to demonstrate that b-value did not strongly depend on the depth change. It suggests that depth dependency expected by Scholz 2016 is not obvious. Font size was enlarged in the revised version.

L102: How do you calculate M_c ?

We adopted M_c calculated by ZMAP.

L106: This should be formatted as an equation as opposed to have the equation in-line. I would also make it clear that this equation is derived from Scholz, 2015 and not the data presented in this work.

We changed the form (L140).

Figure 3: How are the shear and normal stresses normalized? What does it mean to have a negative normal stress? How are the data in Figure 3a /3b segmented? How is Figure 3B subdivided? What do the numbers in the legend in Figure 3D correspond to? Do each one of these F/m lines correspond to the data in Figures 3B-3C. please clarify. Also, it would be useful to point out where M_c is located on these plots and the predicted F/M (i.e., fit) to the data.

Shear and normal stress on a unit Mohr circle can be calculated from normalized deviatoric tensor estimated in this paper and fault plane orientation. We described in the Method section. r and θ can be obtained from normalized shear and normal stresses. After trials of various segmentation, we adequately chose the segment to most exhibit the feature on b-value variation. In Fig.3d, each colored segment corresponds to the r and θ range shown in the legend. We added M_c indicator in the Fig. 3d.

L126: What does the range of radius, r , physically mean? And what does this imply about the relationship shown in Figure 3C. In other words, why does b when r is between 0.8-1, but is independent of normal stress between 0.5-0.5?

Fault plane with " $r=1$ " is that a plane rotates around σ_2 axis with any angle. No effect from σ_2 exists on the plane. Plane with $r < 1.0$ is affected by σ_2 . The diagram implies plane close to $r = 1$ can generate large slip. However, it is not always clear quantitative mechanisms explaining it. Only observational facts that larger slip occurred on the favorable oriented plane exists as described the text. In addition, it cannot be determined in $r = -0.5 - 0.5$ whether b -value is independent of normal stress or not based on present accuracy. However, θ dependency, which also relates to the normal stress, was observed in high r segments.

L122-128: Does this region correspond to a high normal stress? Can you say anything about where/how close the fault is to failure? This would help address the question of whether b -value is low when the fault is critically stressed.

Normal stress depends on the location in the Mohr circle as shown the figure. The interpretation about the b -value distribution is described in the following sentences. We attempt to explain the distribution of b -value based on Coulomb failure criterion. The criticality is also discussed the following sentences. We added an interpretation about this point.

L129: It would be useful to show a mohr-coulomb plot, the failure envelope and the distribution of b -values. Perhaps some of this information is shown Figures 3a-b?

We showed it in the Fig. 4 in the revised version.

L133-136: I do not follow this section. Can you please rephrase/explain what you mean here: "This suggests relatively larger fault slips..."

We rewrote the expression and explained with Fig. 4a about this point.

L144-145: Please explain what you mean here. Where in Figure 3 are the optimal and nonoptimal planes that you are referring to?

We changed the expressions and draw Fig. 4a to explain it.

L146: Why does pore-pressure need to be the causative agent for causing unfavorable

planes to slip?

Pore-pressure declines effective normal stress. Then, strength of the fault decreases and fault slip on unfavorable fault plane easily occurs.

L171-186: What are the spatial and temporal bounds of the catalog used in this study? How many events are in the catalog? 3641? Is the network called "The 2017 Network" or is 2017 simply the year the network was deployed. Please clarify.

We described about this point in the revised version.

L242:-244: Can you provide an example and/or show in Figure A5 how the misfit angle is determined.

We added schematic illustration in Fig. A7 (Fig. A5 in the original version).

L222-265: What are the uncertainties associated with the normalized normal and shear stresses measured here?

The shear and normal stresses were used for categorizing event to part subdivided Mohr circle. As you pointed out, the uncertainties of those effect on number of event belonging the blocks. However, we think the errors expressed in r and θ domain is more direct to consider categorization. Therefore, we added table of error for every area in the Mohr circle as Table. A1.

L269: What do you mean by standard method? Maximum likelihood? How do you determine M_c ?

We use Maximum likelihood method. and M_c was determined by ZMAP.

L277: Why do you exclude events close to the fault plane?

Stress field can be strong heterogeneity at location close to the co-seismic fault plane. It strongly depends on fault geometry. If stress field differ from the estimated one, mis-estimate the categorization of event. Therefore, we need to check result depends on distance from the fault.

Reply to Reviewer #2

Thank you for your comments and suggestions on our article.

We answer written in blue color to your comments as follows.

1. Nevertheless, the link between focal mechanisms, fault stresses and b-values is not described in sufficient detail. The authors should better demonstrate that they are indeed able to robustly constrain focal mechanisms and it would be beneficial to report uncertainties as a function of magnitude. The description of the two different earthquake sequences and respective seismic instrumentation could be more detailed, and the authors should add figures that clearly show which events were selected to invert for stress and b-values (Fig. A3 is not clear enough because it doesn't show the underlying events).

We plotted errors of nodal planes in focal mechanism with magnitude in Fig. A3 to show accuracy of data in both observations. We added description about the observations in the text. We chose all events containing each spatial bin in order to estimate stress tensor. About events selected for stress estimation plotted in a pdf file because of too many figures exist. Normalized shear and normal stress values were calculated only for events that were selected the criteria described in the text. The descriptions about the observations were cited from below papers.

Figure Left) Target area and moment tensor solutions since the 2000 Western Tottori

Earthquake, until Apr 2018, recorded by F-net, NIED. The beach balls show the moment tensor solutions plotted on the lower hemisphere. Crosses show Hi-net and NIED stations used in this study. Triangle indicates an active volcano. The red star indicates the epicenter of the main shock. Dotted lines indicate prefectural boundaries in the region. Right) Station distribution of the “0.1 Manten” dense seismic observations in blue circles. The location of the right map corresponds to the dashed rectangle in the left figure. Other symbols are the same as in the left figure (Matsumoto et al., 2020)

The 2000 OBS was described in Shibutani et al. 2005. We used their picked polarity data.

[REDACTED]

Method:

– Explain and show in more detail which events were selected to compute regional stress tensor variations, magnitude distributions and b-values!

We added sentences explaining process. The process are;

- 1) Dividing space by spatial bins with size 0.03 degree in horizontal and 2.5 km in depth.
- 2) stress tensor was estimated by focal mechanism of hypocenter located in each spatial bin.
- 3) Estimating normalized normal and shear stress for an event from the focal mechanism and the stress tensor at the spatial bin where the event was located.
- 4) All available events that stress tensor successfully estimated were plotted in a unit Mohr circle
- 5) Dividing Mohr circle into subarea (i.e. some r and q ranges).
- 6) calculating b-value for each subarea.

– Results from focal mechanism and stress tensor analysis of differently discretized datasets in 2000 and 2017 should be shown as maps. Fig. A3 is a start but the results are difficult to evaluate.

Exemplary stress tensor maps, can be found in the following papers:

Yang, W., & Hauksson, E. (2013). The tectonic crustal stress field and style of faulting along the Pacific North America plate boundary in southern California. *Geophysical Journal International*, 194(1), 100–117.

<https://doi.org/10.1093/gji/ggt113>

Martínez-Garzón, P., Bohnhoff, M., Kwiatek, G., & Dresen, G. (2013). Stress tensor changes related to fluid injection at the Geysers geothermal field, California.

Geophysical Research Letters, 40(11), 2596–2601. <https://doi.org/10.1002/grl.50438>

We draw figure that is plotted eigen vectors and stress ratio with 95% confidence range. The stress tensors that in used to estimate normalized normal and shear stresses were obtained from merged data of the 2000 and 2017 OBSs because merged estimation provides available blocks where stress tensor estimated. For comparing stress tensor inversion results among data from 2000 OBS, 2017 OBS, and 2000 & 2017 OBSs, we plotted separately those results in Fig. A4.

Results:

- How does the resolved stress orientation compare to spatial variations in b-values? It would be nice to show both results in a side-by-side comparison together with a map of the hypocenters.

Because of small amount of data in each spatial block, we could not estimate b-value for each block. Usually, b-value estimation in different stress condition is difficult because of its dependency on stress. Therefore, firstly, we showed b-value depth dependency small, meaning stress level difference is small. And then, we categorized events on the Mohr circle and estimated b-value. Finally, we found the dependency on location on Mohr circle. It is meaningful b-value dependency on location in Mohr circle.

Presentation and Structure:

- The presentation of the results needs substantial improvements. The paper is poorly structured, and a lot of key information are scattered across the main text, Method section and supplement.

We rewrote descriptions and re-organized.

- Display items and descriptions need much improvement. Lack of clarity in writing and figures significantly subtracts from the otherwise well-conceived study and the unique dataset.

We checked items and descriptions and revised them.

Uncertainties:

- Fig. 3 only shows uncertainty in b-value and stress. It would be helpful to also show how well focal mechanism of small events were resolved.

We showed standard error versus magnitude event in Fig. A3

Minor Comments:

L 24: Grammar! First sentence needs to be rewritten.

We checked. (L25)

L 43: explain the term: “inclination of differential stress” (differential stress is not a tensor!)

We changed the expression (L42).

L 44: Decreasing b-values before large earthquakes are rarely observed. Note that the provided citation (Smith 1981) argued for an increasing in b-value before moderate earthquakes in New Zealand. These statements need to be much more nuanced to accurately portrait the current observations.

We sited the article (L45).

L 58: rewrite!

Yes.

L 68: Grammar

We checked

L 72: rewrite!

Yes.

L 90: Describe the seismic network and type of instruments in more detail!

We added the description. (L112 – 121)

L 92: Explain the two different earthquake sequences in 2000 and 2017/18 in more detail. The point was quite confusing because it was unclear what earthquake sequence was used to derive the primary observations and results.

We made them clear in revised version.

Fig.1: Caption: Dayson volcano is not in the “lower left”

We changed the expression.

Writing and grammar need to be improved throughout.

We asked professional English editing service and checked the document.

REVIEWER COMMENTS

Reviewer #1 (Remarks to the Author):

The authors have addressed all of my comments and concerns from the previous revision. I have a few minor comments below; once addressed I believe the manuscript is suitable for publication.

L83-85 The coulomb failure criteria should be expressed as a formula.

L133-134: Link should be in the supplement or acknowledgements. Not sure that it belongs within the text of the manuscript.

Reviewer #2 (Remarks to the Author):

Overall, the authors provided some additional clarifications, in particular with respect to focal mechanism and stress inversions. Nevertheless, there is a lot of room for improvements. Figure 2 and 3 look the same as before, except for larger legends. Writing and grammar would benefit from more thorough revisions and there is a lack of clarity in describing the utilized datasets (see below). I recommend moderate revisions.

Detailed Comments:

The Gutenberg-Richter formulation on L37 is not a power-law! I already pointed this out in my previous review. The equation would have to be written with earthquake frequency as a function of seismic moment.

L44: What does “temporal increment of the b-value” mean?

L80-L82: Needs to be rewritten - work repetition.

L82-83: repetition

L86: stick-slip or unstable and stable sliding

L 109: Compare the sentence in the manuscript: “we selected the aftershock activity of the 2000 Western Tottori Earthquake of magnitude 7.3”, to the authors statement in the response letter: “stresses were obtained from merged data of the 2000 and 2017 OBSs” and “comparing stress tensor inversion results among data from 2000 OBS, 2017 OBS, and 2000 & 2017 OBSs”, and “We discuss b-value dependency on fault strength by using aftershock activity of 2000 Western Tottori earthquake.”

It is still not clear to me what datasets were used to generate which figures. Note that the Result section and response letter heavily focuses on aftershock sequences while the rest of the paper focuses on stress and strength variations before large earthquake.

I would recommend including a table of the different records of the 2000 and 2017 seismicity, including mainshock origin times and magnitudes, time windows used for the analyses and whether the datasets were used to compute focal mechanisms of b-values. In addition, each figure should clearly describe which dataset is shown.

L 140: Equations should be numbered. Why are there two constants for the slope in this linear relationship? I am also not following the argument. It would be more insightful to show whether there are significant depth-variations in your b-values or not?

L 162: Grammar!

L 210ff: Concluding paragraph should be rephrased. I don't think this study has any implications for temporal b-value variations. The authors are investigating aftershock sequences, so I am not sure what the implications for the seismic cycle are. "the b-value decreases toward the end of the seismic cycle". Was that really observed? Which figure is showing the decrease in b-value before the 2000 and 2017 events?

Reply for comments from reviewers.

To Reviewer #1:

Thank you for your comments. We have revised to follow your comments (reply is indicated by blue colour)

L83-85 The coulomb failure criteria should be expressed as a formula.

We wrote this as a formula.

L133-134: Link should be in the supplement or acknowledgements. Not sure that it belongs within the text of the manuscript.

We delete the link and show in the code availability.

To Reviewer #2:

Thank you for giving us valuable comments. We have revised our manuscript based on your suggestions. The replies are listed below by blue letters. We hope our revision make clear your concerns.

Overall, the authors provided some additional clarifications, in particular with respect to focal mechanism and stress inversions. Nevertheless, there is a lot of room for improvements. Figure 2 and 3 look the same as before, except for larger legends. Writing and grammar would benefit from more thorough revisions and there is a lack of clarify in describing the utilized datasets (see below). I recommend moderate revisions.

We re-organized to describe our intended meaning clearly.

Figure 2. We showed event distribution for estimating b-value (only for 2017 OBS) and co-seismic fault. In this figure, we would like to show no significant b-value difference between depth ranges. Therefore, we divided hypocenter distribution into two depth ranges ($z < 7.5$ km and $z \geq 7.5$ km).

Figure 3. We moved Fig. 3d to Fig. 4 to show the distribution clearly.

Detailed Comments:

The Gutenberg-Richter formulation on L37 is not a power-law! I already pointed this out in my previous review. The equation would have to be written with earthquake frequency as a function of seismic moment.

We corrected expression. Thank you

L44: What does "temporal increment of the b-value" mean?

The higher b-value before large earthquake and back to lower one after the occurrence.

We revised the expression.

L80-L82: Needs to be rewritten - work repetition.

L82-83: repetition

We changed the expression. We expect this change is significant to reflect your intended meaning.

L86: stick-slip or unstable and stable sliding

We corrected. Thank you

L 109: Compare the sentence in the manuscript: “we selected the aftershock activity of the 2000 Western Tottori Earthquake of magnitude 7.3”, to the authors statement in the response letter: “stresses were obtained from merged data of the 2000 and 2017 OBSs” and “comparing stress tensor inversion results among data from 2000 OBS, 2017 OBS, and 2000 & 2017 OBSs”, and “We discuss b-value dependency on fault strength by using aftershock activity of 2000 Western Tottori earthquake.”

We changed and added description about data treatment in order to clarify the usage of dataset.

It is still not clear to me what datasets were used to generate which figures. Note that the Result section and response letter heavily focuses on aftershock sequences while the rest of the paper focuses on stress and strength variations before large earthquake.

I would recommend including a table of the different records of the 2000 and 2017 seismicity, including mainshock origin times and magnitudes, time windows used for the analyses and whether the datasets were used to compute focal mechanisms of b-values. In addition, each figure should clearly describe which dataset is shown.

Thank you for your suggestions.

We added sentences and supplementary Table 1 that was listed information about the

mainshock and two observations. We also added Supplementary figure 1 to show time sequence the main and aftershock activity by using JMA catalogue because we do not have continuous record of seismicity.

L 140: Equations should be numbered. Why are there two constants for the slope in this linear relationship? I am also not following the argument. It would be more insightful to show whether there are significant depth-variations in your b-values or not?

We numbered the equation.

The two constants were reflected original formula by Scholz. One (0.0012) is a constant in his paper. Another (20) is term expressing differential stress per depth in km (20 MPa/km).

We used average depths (5 and 10 km) to calculate empirically expected b-value. For 5km, $b = 1.23 - 0.0012 \times 20 \times 5$.

For 10km, $b = 1.23 - 0.0012 \times 20 \times 10$.

The difference between the b-values is $0.0012 \times 20 \times 5 = 0.12$. Therefore, depth difference between two depth range is approximately 0.1.

We revised expression that show above the calculation.

L 162: Grammar!

We changed expression as;

The b-value is significantly smaller at $90^\circ > \theta > 50^\circ$ with $r > 0.8$ compared to the other ranges.

L 210ff: Concluding paragraph should be rephrased. I don't think this study has any implications for temporal b-value variations. The authors are investigating aftershock sequences, so I am not sure what the implications for the seismic cycle are. "the b-value decreases toward the end of the seismic cycle". Was that really observed? Which figure is showing the decrease in b-value before the 2000 and 2017 events?

As you pointed out, there is no study that reported the proximity change. It is expectation for further observation research. We would like to argue importance of monitoring for strength dependency of b-value. However, we showed that b-value decline found by a b-value estimation from a certain earthquake catalogue can be caused by changing to high proximity to fault failure. High proximity condition is expected at the end of seismic cycle so that we would like to describe that b-value might decrease.

However, present expression was unclear to mentioned above. Therefore, we changed the expression.